# Ideas and perspectives: How sediment archives can improve model projections of marine ecosystem change

Isabell Hochfeld[1], Ben A. Ward[2], Anke Kremp[3], Juliane Romahn[4,5,6], Alexandra Schmidt[7,8], Miklós Bálint[4,5,6], Lutz Becks[9], Jérôme Kaiser[3], Helge W. Arz[3], Sarah Bolius[3], Laura S. Epp[7], Markus Pfenninger[4], Christopher A. Klausmeier[10], Elena Litchman[10], and Jana Hinners[11]

[1]Institute of Marine Ecosystem and Fishery Science, University of Hamburg, Germany
[2]School of Ocean & Earth Science, University of Southampton Waterfront Campus, Southampton, United Kingdom
[3]Leibniz Institute for Baltic Sea Research Warnemünde IOW, Rostock, Germany
[4]Senckenberg Society for Nature Research, Frankfurt, Germany
[5]LOEWE Centre for Translational Biodiversity Genomics, Frankfurt, Germany
[6]Institute of Insect Biotechnology, Justus-Liebig University, Gießen, Germany
[7]Environmental Genomics, Department of Biology, University of Konstanz, Germany
[8]Max Planck Institute of Animal Behavior, Radolfzell, Germany
[9]Aquatic Ecology and Evolution, Department of Biology, University of Konstanz, Germany
[10]Kellogg Biological Station, Michigan State University, U.S.A.
[11]Hemholtz-Center Hereon, Geesthacht, Germany

*Correspondence to:* Isabell Hochfeld and Jana Hinners (isabell.hochfeld@uni-hamurg.de, jana.hinners@hereon.de)

**Abstract.** Global warming is a major threat to marine biodiversity and ecosystem functioning, with consequences that are yet largely unknown. To frame these consequences, we need to understand how marine ecosystems respond to warming and related environmental changes. Ecosystem models have proven to be a valuable tool in this respect, but their projections vary considerably. A major limitation in current ecosystem models may be that they largely ignore evolutionary processes, which nonetheless can be relevant on the simulated time scales. In addition, ecosystem models are usually fit to contemporary data and used predictively afterwards, without further validation that they are equally applicable to past (and by inference, future) scenarios. A promising approach to validate evolutionary ecosystem models are biological archives such as natural sediments, which archive long-term ecosystem changes. Since the ecosystem changes present in sediment records are affected by evolution, evolution needs to be represented in ecosystem models not only to realistically simulate the future but also the sediment record itself. The sediment record, in turn, can provide the required

constraints on long-term evolutionary changes, along with information on past environmental conditions, biodiversity, and relative abundances of taxa. Here, we present a framework to make use of such information to validate evolutionary ecosystem models and improve model projections of future ecosystem changes. Using the example of phytoplankton, key players in marine systems, we review existing literature and discuss (I) which data can be derived from ancient sedimentary archives, (II) how we can integrate these data into evolutionary ecosystem models to improve their projections of climate-driven ecosystem changes, and (III) future perspectives and aspects that remain challenging.

# 1 Introduction

Driven by the reality of global warming as a major threat to marine biodiversity and ecosystem functioning, ecosystem models are increasingly used to estimate future changes in marine ecosystems. However, projected changes differ notably between models (Laufkötter et al., 2015, 2016; Tittensor et al., 2021), which may be due to the fact that evolutionary processes are generally neglected, even though they can be of great importance on the simulated time scales (Irwin et al., 2015; Jin and Agustí, 2018). The reliability of current model projections therefore remains questionable. Here, we propose to use data from sediment archives to validate evolutionary ecosystem models before using them predictively and discuss how this approach can improve model projections.

Compared with the period 1850–1900, global surface temperature has already increased by 1.25 °C and, under the most extreme emissions scenario, is expected to increase by up to a further 3.5 °C by the end of the century (IPCC scenario SSP5-8.5, Allan et al., (2021). The current state of warming has already caused changes in marine communities (Peer and Miller, 2014; Poloczanska et al., 2013; Wasmund et al., 2019), which perform ecosystem functions that are vital to human societies, including food production (Hollowed et al., 2013) and carbon sequestration (Hain et al., 2014). The response of these ecosystem services under ongoing global warming remains subject to great uncertainty, and there is a real but unknown risk of positive feedbacks, irreversible tipping points, and ecosystem collapse (Lenton et al., 2008). For example, IPCC models systematically underestimate the decline in Arctic summer sea ice, which suggests that a tipping point may already have or may soon be passed (Lenton et al.,

2008; Stroeve et al., 2007). Passing this tipping point can cause strong nonlinear and long-term changes in the system, such as amplified warming and sea ice loss through a positive albedo feedback, and, in turn, major ecosystem change (Holland et al., 2006; Lenton et al., 2008).

Dynamic ecosystem models currently represent the best tool to understand complex feedbacks between evolving ecosystems and their environment, but it is a considerable challenge to develop models that would apply equally well to past, present, and future scenarios. Despite their great potential, current models project diverging changes in ecosystem functions like carbon cycling and net primary production (Laufkötter et al., 2015; 2016; Tittensor et al., 2021). For example, a recent coupled model intercomparison showed a variability of up to ± 10 % in projected global net primary production, depending on the ecosystem model and earth system forcing used (Tittensor et al., 2021). This disagreement between models illustrates how critical it is to assess the validity of model projections.

To improve model projections, we need (I) to verify that the most relevant processes are considered and (II) to validate projections with long-term data. Regarding (I), current ecosystem models largely ignore a crucial process that can influence ecosystem responses to environmental changes on perennial time scales – evolutionary adaptation (Hattich et al., 2024; Irwin et al., 2015; O'Donnell et al., 2018). Some ecosystem models already consider adaptation, but only a small number have been compared to empirical data, both from experiments (Denman, 2017) and from sediment archives (Gibbs et al., 2020; Hinners et al., 2019). So far, however, testing against data has not been used to improve projections made by these models. With respect to (II), both experiments and marine monitoring studies cannot account for environmental changes on longer than decadal time scales, while experiments can hardly capture the complexity of real ecosystems. Natural archives such as sediments, however, allow the reconstruction of long-term ecosystem responses to past environmental changes (Capo et al., 2021; Ellegaard et al., 2020). Sediments preserve abiotic and biotic environmental proxies (Hillaire-Marcel and De Vernal, 2007), other organismal remains such as DNA (Alsos et al., 2022; Monchamp et al., 2016; Zimmermann et al., 2023), and dormant resting stages. Dormant resting stages, which are formed by many phytoplankton and zooplankton taxa, are typically characterized by resistant, thick walls and the capacity to endure long phases of physiological dormancy (Ellegaard and Ribeiro, 2018). Similar to seeds, these resting stages can be resurrected and used for experiments (Bennington et al., 1991; Hinners et al., 2017; Isanta-Navarro et al., 2021). Since sediments can be dated, we can use the preserved

information to derive long-term time series on past environmental conditions, biodiversity, relative taxa abundance, and adaptive changes in (functional) traits. Thus, sediment archives are well suited to constrain the long-term evolutionary changes needed to validate evolutionary ecosystem models. Evolution, in turn, needs to be represented in ecosystem models to simulate its impact on the sediment record.

Here, we discuss how we can use data from sediment archives to improve evolutionary ecosystem models and their projections of marine ecosystem change. Our approach focuses on phytoplankton, key players in marine ecosystems and respective models. Phytoplankton account for about half of global photosynthesis (Field et al., 1998), are the basis of the marine food web (Fenchel, 1988), represent an important component of biogeochemical cycles (Hutchins and Fu, 2017), and can even influence ocean physics (Hense, 2007; Sathyendranath et al., 1991). In addition, the large population sizes and short generation times of phytoplankton allow them to adapt quickly to changing environmental conditions (Aranguren-Gassis et al., 2019; Hattich et al., 2024; O'Donnell et al., 2018). All these factors, together with their long-lived dormant resting stages (Delebecq et al., 2020; Sanyal et al., 2022), make phytoplankton ideal model organisms for the approach we present here. Based on existing literature, we discuss which data we can obtain from sediment archives, how we can use these data to improve evolutionary ecosystem models and their projections, and remaining challenges and future perspectives.

# 2 Sediment archives – understanding phytoplankton responses to environmental change

Sediment archives provide information on past ecosystem status, including both environmental and biological data (Fig. 1). Such data can be used to constrain evolutionary ecosystem models.

123

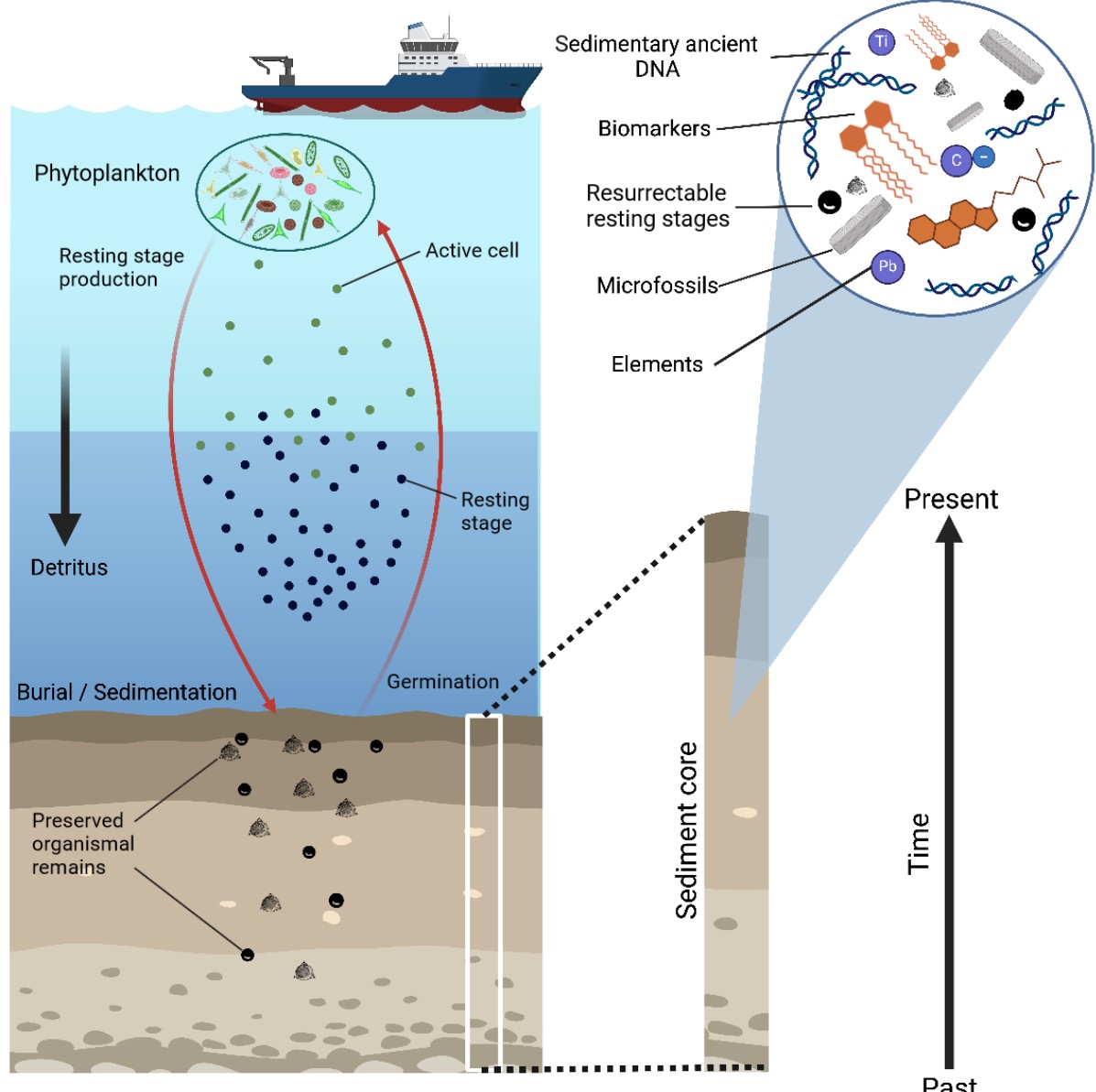

**Figure 1:** Overview of different types of data (environmental and biological) that can be obtained from sediment archives. **Left:** Schematic showing the deposition of organismal remains in the sediment. Red arrows indicate the production of dormant resting stages (thick-walled cells that can endure long phases of physiological dormancy), their deposition in the sediment, and the germination of resting stages from the sediment. The black arrow represents sinking of dead organic matter (detritus) to the seafloor. Preserved organismal remains, a mixture of resting stages and detritus, are shown in the sediment. **Right:** Close-up of the sediment core showing different types of data that can be obtained. The figure was created with BioRender.com.

132

## 2.1 Dating sediment archives

Before working with sediment archives, the sediments must be dated accurately to obtain a well-constrained relationship between age and sediment depth, a so-called age model. Common sediment dating methods include radiocarbon dating, lead isotope dating, and event stratigraphy. Radiocarbon ($^{14}$C) dating is based on $^{14}$C half-life. Determining the amount of radioactive $^{14}$C relative to the $^{12}$C stable isotope allows estimating age ca. 50,000 years back in time (Hajdas et al., 2021). After 1950, radiocarbon dating is not applicable anymore due to the radiocarbon added artificially to the atmosphere by nuclear bomb tests. Therefore, sediments deposited after 1950 are dated using different methods such as lead isotope ($^{210}$Pb) dating and event stratigraphy. While the $^{210}$Pb dating approach is based on the half-life of atmospheric $^{210}$Pb (Appleby, 2001), event stratigraphy is based on the detection of specific events such as nuclear bomb tests, volcanic eruptions, and other distinct anthropogenic impacts that are registered in, for example, chemical parameters of the sediments (Hancock et al., 2011; Lowe and Alloway, 2015). By combining all the dating methods mentioned above, it is possible to obtain robust age models of sediment cores over the last ca. 50,000 years. Other stratigraphic methods, e.g., oxygen isotope stratigraphy, biostratigraphy, or paleomagnetic stratigraphy are applied to date older sediments deposited in aquatic environments (Bradley, 1999).

## 2.2 Environmental data

Abiotic and biotic proxies, or indicators, preserved in sediment archives allow the reconstruction of physicochemical characteristics of past marine and limnic environments. For example, surface salinity can be estimated using lipids (alkenones) produced by micro-phytoplankton species of the order Isochrysidales (Kaiser et al., 2017; Medlin et al., 2008; Rosell-Melé, 1998). Some trace metals and their isotopes are indicators for past suboxic to euxinic conditions in the water column and/or the sediments (Brumsack, 2006; Dellwig et al., 2019). Relative assemblages of microfossils (e.g., resting stages of dinoflagellates, silica frustules of diatoms, calcareous shells of foraminifera) and their shell geochemistry provide important information not only on salinity, but also on pH, trophic state, and temperature, and are therefore powerful proxies (Cléroux et al., 2008; Hillaire-Marcel and De Vernal, 2007; Lear et al., 2002). Organic indexes based on biomarkers, e.g., alkenones ($U^K_{37}$, Prahl et al., (1988)) or other membrane lipids derived from archaea ($TEX_{86}$, Schouten et al., (2013)) can be used to reconstruct surface and subsurface temperature. These and many other physical methods,

biological proxies, and geochemical tracers find their diverse applications in paleoceanography
(Hillaire-Marcel and De Vernal, 2007).
Proxy-based reconstructions must be considered carefully as they may be biased due to
preservation/degradation and influenced by local-to-regional environments. Using a
multiproxy approach and calibration depending on the environment is important for reliable
reconstructions. Reconstructed environmental conditions of the past can then be used as forcing
for ecosystem models.

## 173 2.3 Biological data

Apart from information on environmental conditions, sediment archives provide a wide variety
of biological information, such as biodiversity, relative taxa abundance, and trait adaptation.
This biological information is valuable for validating evolutionary ecosystem models.

### 178 2.3.1 Microfossils

Traditionally, the focus of research on sediment archives has been on fossilized plankton
remains. Fossil phytoplankton communities only represent species that consist of stable
mineral structures (e.g., of silica or carbonate) or contain specific fossilizable molecules such
as sporopollenin. Among dinoflagellates, only a fraction of the community produces resting
cysts (Limoges et al., 2020; Van Nieuwenhove et al., 2020), which are preserved over time and
can be used for quantitative paleoecological reconstructions and biostratigraphy. Diatoms, on
the other hand, are well-represented in the fossil record due to their resistant silica frustules
with their diverse species-specific structures (Weckström, 2006). Some filamentous
cyanobacteria also produce resistant resting stages, akinetes, constituting long-term records in
brackish-marine and lake sediments (Wood et al., 2021). In lakes, chrysophyte cysts can
provide long-term records that reveal group-specific phytoplankton dynamics over long time
scales (Korkonen et al., 2017). While microfossil data provide continuous (semi-)quantitative
records of the relative biomasses of the represented taxa and larger taxonomic groups (e.g.,
cyanobacteria, diatoms, and dinoflagellates) over geological time scales, their biodiversity
information is limited. Only a fraction of taxa within the phytoplankton community is usually
represented in the fossil record, and therefore, respective data are likely biased (Bálint et al.,
2018). Nevertheless, for those taxa that are suitable and sufficiently represented, highly
informative demographic data can be generated from microfossil and resting stage records
(Cermeño et al., 2013; Kremp et al., 2018; Matul et al., 2018). Furthermore, data on the
temporal distribution of larger taxonomic groups as obtained from microfossil records can
provide general information on trait composition and function of phytoplankton communities
through taxonomic identity (Blank and Sánchez-Baracaldo, 2010).

## 2.3.2 Sedimentary ancient DNA
To capture biodiversity dynamics of phytoplankton through time, recent advances in
sedimentary ancient DNA approaches can increase taxonomic coverage and resolution. DNA
can be preserved for thousands or even millions of years in natural sedimentary archives, such
as limnic sediments (Capo et al., 2021; Clarke et al., 2019), marine sediments (Armbrecht et
al., 2022; Coolen et al., 2009, 2013), paleosols (Frindte et al., 2020; Semenov et al., 2020) and
permafrost (Kjær et al., 2022; Willerslev et al., 2003). Compared to microfossils, a distinctive
characteristic of ancient DNA data lies in their capability for broad taxonomic coverage.
Because every organism contains DNA and the differences between species are defined by
their DNA, in theory, DNA could be used to identify any organism that became part of the
sediment deposits (Bálint et al., 2018). Establishing relative abundances of organisms from
their DNA record is challenging though. While fossilized remains of certain phytoplankton
taxa can inform us about cell counts, DNA records can be informative about copy numbers of
particular genes (Mejbel et al., 2021). Since gene copy numbers can vary by orders of
magnitude across species, inferences about abundance can be challenging with methods that
target many taxa at once (Vasselon et al., 2018). If the focus is on a narrow set of taxa, gene
copy number information provided by quantitative analyses (qPCR or ddPCR) might be more
readily translated into abundance information, especially if the range of gene copy numbers per
cell can be estimated for the focal species (Godhe et al., 2008). This approach potentially allows
one to obtain demographic information on a targeted taxon in specific sediment horizons.

## 2.3.3 Resurrectable dormant resting stages

Living sediment archives are formed by temporal deposits of dormant resting stages, which can be obtained from phytoplankton that produce long-lived resting cells/seeds (Härnström et al., 2011; Hinners et al., 2017), but are also represented in organisms such as marine microbes (Lomstein et al., 2012; Wörmer et al., 2019), terrestrial plants (McGraw et al., 1991; Sallon et al., 2008), and zooplankton (Kerfoot and Weider, 2004; Pauwels et al., 2010).

Laminated sediments, which form under anoxic conditions due to the absence of mixing by benthic fauna, contain distinct age cohorts of dormant or quiescent phytoplankton (Ellegaard et al., 2017). Such resting stages can germinate when exposed to oxygen, and cells start growing when suitable temperature, light, and nutrient conditions are provided. A number of studies have demonstrated the "resurrection" potential of different phytoplankton taxa after extended periods of resting, ranging from decades to millennia (Bolius et al., 2025; Härnström et al., 2011; Medwed et al., 2024; Sanyal et al., 2022).

Phytoplankton strains that have been re-established from germinated resting stages of different temporal sediment layers can be characterized pheno- and genotypically (Härnström et al., 2011; Hinners et al., 2017; Medwed et al., 2024). Comparison of trait values among temporal cohorts provides information on trait changes, their rates of change, and the mechanisms behind those changes (Hattich et al., 2024). The general adaptive potential of phytoplankton has already been documented in resurrection experiments focusing on different phytoplankton functional traits, such as temperature-dependent growth and nutrient uptake (Hattich et al., 2024), resting stage formation (Hinners et al., 2017), and toxicity (Wood et al., 2021). The structure of ecosystem models is often based on such phytoplankton functional traits, which describe the dependence of phytoplankton organisms on external factors (e.g., reaction norms), their life cycle (e.g., life cycle transitions), or interactions with other organisms (e.g., cell size, toxicity). To realistically assess the extent of adaptations to environmental changes and to use this information for future ecosystem models, it is therefore important to collect comprehensive information on changes in phytoplankton functional traits using resurrection experiments.

Phenotypic trait data from resurrected cultures can also be linked to their underlying genetic components. A common method for this is represented by genome-wide association studies (GWAS) (Hirschhorn and Daly, 2005; Uffelmann et al., 2021; Visscher et al., 2017). GWAS connect variations in the DNA sequence, known as single nucleotide polymorphisms (SNPs), to a specific trait. To do so, it is necessary to validate the candidate loci identified in

GWAS by experiments that target the phenotypic functionality of these loci (Pfenninger, 2024). GWAS approaches can help to determine if certain functional groups of genes (e.g., those involved in oxidation or $CO_2$ fixation) were selected for or lost over time. In addition, GWAS approaches can help to determine whether the traits of interest are polygenic and can thus be adequately modeled as continuous quantitative traits. The success of this method depends on several factors, including the quality of the phenotypic data and the accuracy of the genetic data. Moreover, when co-analyzing compositional DNA data and quantitative phenotypic data, careful transformation, standardization, and the use of specialized statistical methods are essential to avoid misleading conclusions (Gloor et al., 2017).

# 3 Integration of data from sediment archives into evolutionary ecosystem models

Ecosystem models provide a powerful tool to study the functioning of marine ecosystems and their responses to environmental change. For example, ecosystem models can be used to understand global patterns of phytoplankton diversity (Dutkiewicz et al., 2020; Ward et al., 2012). In addition, they can help to identify potential feedback loops (e.g., between cyanobacteria and their environment (Hense, 2007) and trade-offs (e.g., between phytoplankton diversity and productivity, (Smith et al., 2016)). Finally, ecosystem models can simulate how phytoplankton (and zooplankton) respond to different biotic and abiotic factors, including viruses (Krishna et al., 2024; Weitz et al., 2015), eutrophication (Gustafsson et al., 2012), ocean acidification (Dutkiewicz et al., 2015), and temperature changes (Elliott et al., 2005; Lee et al., 2018).

## 3.1 The neglected role of evolutionary adaptation in ecosystem models

Over the past few years, ecosystem models have been increasingly used to estimate the impact of global warming on marine ecosystems and their functioning. Although the results of such studies are relevant for stakeholders (Intergovernmental Panel on Climate Change (IPCC),

(Allan et al., 2021)), current models vary widely in their formulations and predictions, with some models even disagreeing on the direction of change (Laufkötter et al., 2015, 2016; Tittensor et al., 2021). We argue that a major uncertainty in current models is that they do not account for the high adaptive potential of phytoplankton.

Experiments and observations demonstrated that phytoplankton adaptation can be important on multi-year time scales (Aranguren-Gassis et al., 2019; Hattich et al., 2024; O'Donnell et al., 2018) and may hence alter predicted ecosystem changes notably (Ward et al., 2019). Indeed, a recent modeling study revealed that adaptation can significantly reduce simulated warming-related changes in phytoplankton phenology and relative taxa abundance (Hochfeld and Hinners, 2024a). Changes in phenology and relative taxa abundance, in turn, may have a direct impact on ecosystem functioning (Edwards and Richardson, 2004; Hochfeld and Hinners, 2024b; Litchman et al., 2015). To conclude, it is becoming increasingly clear that adaptation cannot be neglected in global warming simulations, putting current models and their predictive ability into question.

Evolutionary adaptation can be integrated into ecosystem models by allowing for one or more phytoplankton traits to change on intergenerational time scales. In case of changing temperature, for example, phytoplankton thermal adaptation can be represented with an evolvable optimum temperature for growth (Beckmann et al., 2019; Kremer and Klausmeier, 2017). Different approaches exist to integrate adaptation into ecosystem models, with the most suitable approach depending on the research question, see Box 1 for details. Overviews can also be found in Beckmann et al., (2019) and Klausmeier et al., (2020b). However, integrating adaptation into ecosystem models brings new challenges, such as identifying the relevant traits and the associated limits and trade-offs (O'Donnell et al., 2018; Ward et al., 2019). One approach to obtain the necessary evolutionary information is represented by evolution experiments, in which populations are kept under controlled environmental conditions for long periods of time (weeks, months, or even years) in order to measure their evolutionary adaptation to these environmental conditions (Hinners et al., 2024; O'Donnell et al., 2018; Schaum et al., 2017). Since such experiments can neither replicate the complexity of real ecosystems nor long-term environmental change, we argue that sediment archives as "natural evolution experiments" represent a valuable complementary source of information, which we explain further below.

## Box 1: Evolution in ecosystem models

Different approaches exist to integrate evolutionary processes into ecosystem models. The approaches differ in their biological complexity and their computational efficiency, which makes them suitable for different applications (see Table 1). Thus, the mutational algorithm should be chosen depending on the research question.

Individual-based models (IBMs) provide the most complex and most realistic representation of evolutionary adaptation. IBMs simulate individual cells with their individual phenotypic trait values (Beckmann et al., 2019; Clark et al., 2011; Collins, 2016); a prominent approach was developed by (Beckmann et al., 2019), who assume that individual cells take up nutrients, grow, divide, and die. Evolution is implemented through random mutations, which occur every several hundred cell divisions (Lenski and Travisano, 1994). The new trait value of the mutated daughter cell is sampled randomly from a normal distribution, which is centered at the parental trait value with a prescribed standard deviation. Thus, small mutations are much more likely than large mutations, while beneficial and deteriorating mutations are equally likely. With natural selection acting over time, the cells with the highest fitness that need the lowest time to divide again become dominant. Maladapted cells, on the contrary, die off eventually if their doubling rate is lower than stochastic losses like mortality and grazing, leading to progressive adaptation of the entire population.

Simulating natural populations with millions of individuals requires a lot of computational power. To reduce computational power demands, identical cells are often grouped into one model variable (Hellweger and Bucci, 2009), a so-called super-individual or agent. Due to the larger number of cells combined into one agent, mutations are assumed to occur more frequently but with a smaller step size (i.e., standard deviation) (Beckmann et al., 2019). Even though agent-based models require less computational power than IBMs, they are usually only integrated into 0-dimensional model environments.

A computationally much more efficient approach is provided by continuous trait-diffusion models (Chen and Smith, 2018; Le Gland et al., 2021; Smith et al., 2016), which do not resolve individual phenotypes as discrete entities. Instead, they compute the mean and variance of a trait for the entire phytoplankton compartment. Trait variance represents the diversity of ecotypes, with mutations adding new variance that can be selected. Selection, in this context, means that the mean trait increases when higher trait values are associated with

higher net growth rates. Due to their low computational power requirements, continuous trait-diffusion models can be integrated into 1D or 3D model environments. Their simplistic representation of evolution, however, comes at a price: While the shape of the trait distribution must be prescribed, trade-offs between traits are difficult to implement, and evolutionary branching cannot be represented.

Multi-compartment trait-diffusion models can be described as the discretized version of continuous trait-diffusion models. The phytoplankton compartment is divided into multiple sub-compartments that differ slightly in one or more trait values (Hinners et al., 2019; Kremer and Klausmeier, 2013; Sauterey et al., 2017). Mutations are implemented as small, non-random fluxes of biomass between the sub-compartments. This approach is of intermediate biological complexity, which is also reflected in computational power demands. Multi-compartment trait-diffusion models require more power than continuous trait-diffusion models but less than IBMs or agent-based models and can hence be integrated into 1D model environments.

|  | IBM/Agent-based model | Continuous model | Multi-compartment model |
|---|---|---|---|
| Mutations | randomly drawn from normal distribution | small mutations | small mutations |
| Biological complexity | high | low | intermediate |
| Computational power requirements | high | low | intermediate |
| Applicability | 0D models | 0D to 1D to 3D models | 0D to 1D models |

**Table 1:** Summary of evolutionary modeling approaches and their applicability.

# 3.2 Building an evolutionary ecosystem model including data from sediment archives

It is a considerable challenge to develop ecosystem models that can be applied equally well to past, present, and future scenarios. Most state-of-the-art ecosystem models are developed in a

two-step process that comprises the definition of prior estimates of parameter values (initialization) and the iterative fit to contemporary observations through parameter adjustment (calibration). We argue that this approach relies too heavily on how an ecosystem is structured in the present, so that models may no longer be applicable when ecosystem structure has changed in the future. To avoid these problems, models should represent fundamental processes that apply more generally instead of being tailored to a specific ecosystem. The general applicability of a model can be tested with an additional step during model development, validation, which makes use of data from sediment archives. While validation is already common for atmosphere and ocean models (Hollingsworth, 1994; Tonani et al., 2015), it has been largely ignored by the ecosystem modeling community. A recent study presented a validation approach for non-evolutionary terrestrial ecosystem models, which is mainly based on plant remains (Alsos et al., 2024). Our approach focuses on phytoplankton, key players in marine ecosystems and respective models. Due to the high evolutionary potential of phytoplankton, we additionally consider evolutionary processes.

The development approach for evolutionary ecosystem models that we propose here comprises three different steps: initialization, calibration, and validation (Fig. 2). Both initialization and calibration are performed using contemporary data, while validation requires data from sediment archives. Only when all three steps have been completed should a model be used to simulate future ecosystem changes.

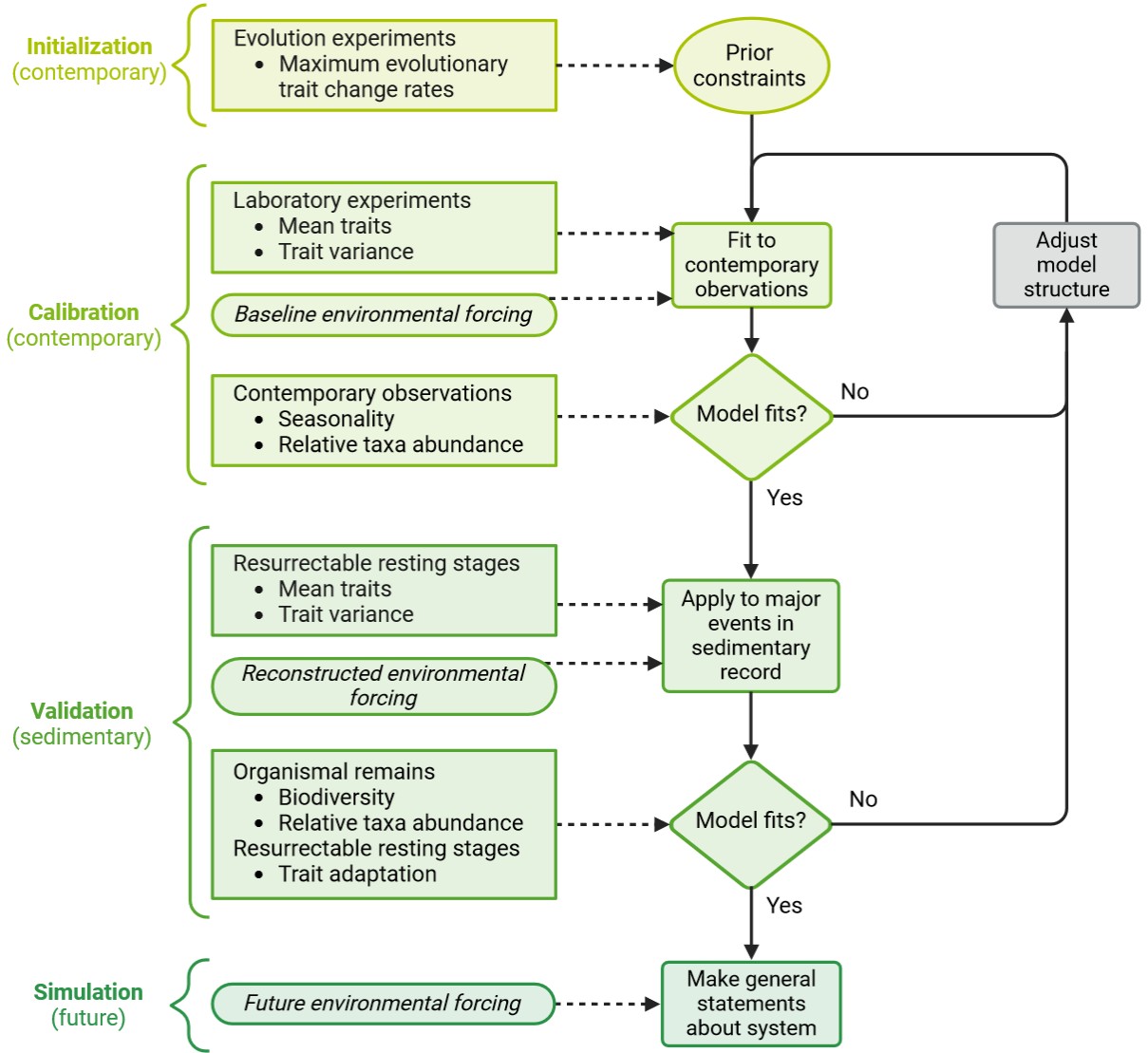

392

**Figure 2:** Conceptual framework for the development of an evolutionary ecosystem model that can be applied equally well to past, present, and future scenarios. Shown are the three different steps of model development (initialization, calibration, validation), the following application of the model (simulation), and the data required for each step. The figure was created with BioRender.com.

397

Initialization requires prior estimates of parameter values that need to be valid regardless of the simulated environmental scenario. Such parameters include constraints on adaptation, such as maximum evolutionary trait change rates, which are, however, difficult to assess. For example, evolutionary trait change rates can be assessed by comparing ancestral trait values to those from populations evolved in a new environment for a specific time after accounting for plastic responses (Collins and Bell, 2004; Hutchins et al., 2015; Listmann et al.,

2016). In addition, it is possible to measure changes in fitness proxies, most commonly
population growth rate or lineage competitive ability (Elena and Lenski, 2003). However,
interpretation is not straightforward since the relationship between fitness and its proxies may
change over time (Collins et al., 2020). Finally, genetic mutation rates can be estimated via
genome sequencing (e.g., (Krasovec et al., 2019), but genetic mutation does not necessarily
translate into trait changes. While functional traits may depend on multiple genes (epistasis),
one gene may affect multiple traits (pleiotropy) (Lässig et al., 2017; Østman et al., 2012; Tyler
et al., 2009). To conclude, evolutionary trait change rates can only be assessed roughly and
require further adjustment in the next steps of model development.

413       The goal of model calibration is to fine tune the model parameters and the model
structure until the model reproduces contemporary observations. To do so, initial values for
mean traits and trait variance are required (Fig. 2). These parameters can be measured in the
laboratory for recently sampled organisms (Lehtimaki et al., 1997; Vincent and Silvester,
1979). The model is then forced with a baseline environmental forcing, usually a steady
seasonal forcing that represents present-day conditions. Using this forcing, the model is run
until it reaches a steady state, where phenology and taxa abundances repeat each season.
Simulated phenology and taxa abundances are then compared to contemporary observations
from seagoing research and remote sensing. If the model does not reproduce the observations,
model parameters and structure are adjusted iteratively until model output and observations
match. When adjusting parameter values, care must be taken to keep all values within a realistic
range, especially when such a range can be constrained by measurements e.g., for traits like
half-saturation constants and maximum growth rates (Eppley et al., 1969; Hinners et al., 2017).
When adjusting the model structure, on the other hand, it should be reconsidered whether the
most relevant processes (e.g., life cycle transitions, mortality, ecological interactions such as
grazing) are included and described realistically.

429       As the final step, model validation aims to test if the model is equally applicable to past,
present, and (by implication) future scenarios by comparing the model output to independent
validation data. We argue that data from sediment archives are ideally suited for validation,
with a contemporarily calibrated model being successfully validated if it can represent major
shifts in community structure and/or function that are present in the sediment record. As a first
step, validation needs initial values for the mean and variance of the relevant traits. These
parameters can be measured in the laboratory for resurrected organisms sampled from the

sediment layer that corresponds to the beginning of the validation period. In addition, environmental conditions during the validation period must be reconstructed to create a forcing for the model. Extreme climate periods such as the Pliocene (5.3-2.6 million years ago), when the global mean surface temperature was ~2°C higher than today and more than 10°C higher at high latitudes (Ballantyne et al., 2010; Haywood et al., 2000; Salzmann et al., 2008), or the last interglacial period (128,000-116,000 years ago, Muhs, 2002; Stirling et al., 1998), when temperatures in the Arctic were up to 5°C higher than today (North Greenland Ice Core Project members, 2004) would be ideally suited to test the model's validity in extreme and changing climates. The simulated biodiversity and relative taxa abundances can then be compared to organismal remains from different sediment layers throughout the validation period. Similarly, simulated trait changes can be compared to results from resurrection experiments, which are performed with organisms from different sediment layers of the validation period. If the contemporarily calibrated model cannot reproduce major events in the sediment record, this implies that the model's structure and parameterization are not general to both contemporary and past systems and should therefore not be used to make predictions.

For example, (Gibbs et al., 2020) used an evolutionary ecosystem model that was parameterized in accordance with contemporary laboratory measurements to reproduce an observed shift in the trophic status of coccolithophores after the end-Cretaceous mass extinction. However, while the model produced an evolutionary response that was qualitatively consistent with the sediment record, the simulated evolutionary response progressed at a rate that was orders of magnitude too fast. This indicates that the model would require further adjustments to allow reproduction of both contemporary and sedimentary data before the model could be used predictively.

While such a model could be recalibrated to fit the past data, we do not recommend this approach, because the ad hoc adjustment of the model parameters does not fix the underlying problem. In addition, calibration is not possible when making predictions. Therefore, instead of recalibrating the model to past data, we advocate refining the model structure to better represent processes that do apply generally, across past, present, and future systems. After recalibration to contemporary data, the refined model could be tested again against past data. Repeating this process iteratively until both contemporary and past data can be reproduced with the same model assures that the model can provide meaningful statements about an ecosystem's possible response to future climate changes.

468        To estimate remaining model uncertainties, the uncertainties in parameter values and

environmental forcing can be used for sensitivity experiments. The extent to which the model
structure influences the results can be assessed by comparing the model results before and after
the structural change.

## 473    3.3. Exemplary implementation

Below, we describe an exemplary implementation of our approach using the following research
question: How did and will the phytoplankton community in the Baltic Sea change during the
past and future 100 years, and what role does thermal adaptation play in these changes?

477        For our exemplary implementation, we assume that sediment cores have already been

collected and analyzed. Thus, it is known which temperature-dependent traits have changed
over the past 100 years before our approach is implemented. In this example, we assume that
an adaptive response in the optimum temperature is evident in the sediment record, as reported
in e.g., Hattich et al. (2024).

482       1.   Ecosystem model setup and initialization:

As a first step, the evolutionary ecosystem model structure is built. The model structure
mathematically describes the dynamics and interactions between components of the Baltic Sea
ecosystem that are considered relevant for answering the research question. In this case, the
model structure includes equations for nutrients, the most relevant phytoplankton functional
groups in the Baltic Sea (dinoflagellates, diatoms, and cyanobacteria), and dead organic matter
(detritus). For each phytoplankton functional group, a growth equation is formulated, including
a limitation function for nutrients. In addition, functions for life cycle transitions (e.g.,
following (Hochfeld and Hinners, 2024a) and mortality are implemented. The respective
formulations and parameters (i.e., half- saturation constants, basic transition rates in life cycle
transition functions, basic mortality rates in mortality functions) are adopted from literature
(e.g., Eppley et al., 1969; Hense and Beckmann, 2006; Hinners et al., 2019). A 1-dimensional
water column model is used for the simulations (e.g., GOTM, Li et al., 2021; Umlauf et al.,
2005), evolution is enabled using a multi-compartment model (e.g., Hinners et al., 2019). The
evolutionary algorithm is initialized with experimentally derived evolutionary trait change
rates for optimum temperature, which can be obtained from literature (e.g., Jin and Agustí,
2018; Listmann et al., 2016).
2.  Model calibration:
For calibration, the model needs to be complemented by temperature limitation functions for
the simulated taxa. These temperature limitation functions are established from the thermal
reaction norms of resurrected resting stages from the uppermost sediment layer, including the
mean and the variance of the optimum temperature. In addition, a baseline temperature forcing
is created to represent recent annual temperature fluctuations in the Baltic Sea, e.g., by using
temperature           data           from           the           Copernicus           database
(https://resources.marine.copernicus.eu/products). The model is then run with the baseline
forcing until a steady state is reached, where the simulated dynamics repeat each season.
Finally, the model output is compared to existing, recent data on the relative abundances and
seasonal dynamics of the simulated phytoplankton taxa (e.g., Hjerne et al., 2019). If the model
does not reproduce the data, both the model structure, i.e., the functions for nutrient limitation,
life cycle transitions, and mortality, as well as the corresponding parameters initialized in step
1 are adjusted until the model output fits the data.
3.  Model validation:
For validation, the model is initialized with mean optimum temperatures and variances from
100 years ago. For this purpose, data from resurrection experiments with resting stages from
the historic sediment layer are used (e.g., Hattich et al., 2024). A hindcast temperature forcing
for the last 100 years is reconstructed from the sedimentary record using biotic proxies such as
microfossils and biomarkers (e.g., Kabel et al., 2012; Wittenborn et al., 2022). After applying
the hindcast forcing to the model, model performance is evaluated based on a biological
sediment assessment of changes in relative taxa abundance (using genetic analyses and the
assessment of resting stages across the sediment, e.g., Kremp et al., 2018; Schmidt et al., 2024),
and changes in the optimum temperatures of these taxa (using resurrection experiments, e.g.,
Hattich et al., 2024). If the model is not able to reproduce changes in relative taxa abundances
and past adaptation of the optimum temperature, its structure needs to be refined. For example,
the simulated evolutionary response may be less pronounced than the one derived from the
sedimentary record. This issue could be fixed by increasing the evolutionary trait change rates
until the model reproduces the observed evolutionary changes, which may result in
unrealistically high evolutionary trait change rates. Instead of simply adjusting parameters, it
may therefore be necessary to consider other hitherto disregarded processes that may exert an
additional selection pressure, for example, a temperature-dependent cyst mortality (Hinners et
al., 2019). If the model reproduces past adaptive changes and taxa abundances with the new
cyst mortality function, model performance is tested again for the baseline environmental
forcing. If the revised model still reproduces the calibration data, it is equally applicable to the
past 100 years and the present and can therefore be used for the intended simulations of future
climate change.

# 4 Challenges and potential of using data from sediment archives for evolutionary ecosystem modeling

Our approach has the potential to increase the informative value of model projections of future
changes in marine ecosystems. However, there are still some challenges associated with it.
A major challenge is posed by the low temporal resolution of sediment records, which
can range from multi-centennial to annual depending on the sedimentation rate (Abrantes et
al., 2005; Maslin et al., 2003). Thus, phenological information is missing even in high-
resolution records, meaning that simulated phenology cannot be validated using data from
sediment archives. Instead, simulated phenology can be validated using monitoring data, which
may go back several decades (Wasmund et al., 2019).
Assuming that evolutionary change in phytoplankton can occur on a decadal time scale
(Irwin et al., 2015), sediments that accumulate at similar or even higher rates, which is usually
the case in lakes and marginal seas, should allow the detection of evolutionary change (based
on resurrection experiments or genetic analyses). At sites with lower sedimentation rates, rates
of evolutionary change may be underestimated. It is therefore recommended to work in
environments with high sedimentation rates (e.g., $\geq 1$ mm/year) to investigate evolutionary
changes. In addition, the age model uncertainty, which increases with the age of the sediments,
can lead to inaccuracies in the determined rates of evolutionary change. Depending on the
depositional environment, preservation rates (decomposition rates) of organic matter vary
greatly (Canfield, 1994; Hedges and Keil, 1995; Wakeham and Canuel, 2006). Preservation is
lowest in oxic environments with low sedimentation rates and highest in anoxic environments
with high sedimentation rates (Canfield, 1994). Differential preservation may also bias
estimates of relative abundance, although structurally similar organic compounds are expected
to be similarly preserved (Wakeham and Canuel, 2006). To determine possible effects of
preservation on the estimates of (relative) abundance and evolutionary changes, it is important
to estimate the content of total organic carbon in the sediment. For the investigation of
evolutionary changes, it is therefore recommended to work in environments with high
sedimentation rates, low age model uncertainty, and hypoxic to anoxic conditions. Among
others, the Baltic Sea, the Black Sea, the Cariaco Basin, and many lakes represent such suitable
environments.
Dormant resting stages that have been preserved in the sediment record and could be
revived for experiments may not be representative of the entire population at the time of
deposition, and therefore may not be representative of its traits. However, assuming that the
fittest individuals of a population were most abundant in the past and hence are most abundant
in the sediment, we should be able to measure representative mean trait values for the
population. Nevertheless, we cannot rule out that storage in the sediment may have distorted
the measurable characteristics of a population. To obtain reliable estimates of trait variance,
experimental studies on phytoplankton traits should therefore aim to characterize as many
strains as possible, e.g., using high-throughput methods (Argyle et al., 2021). Proxies for the
reconstruction of environmental conditions and DNA can also suffer from
preservation/degradation biases and are therefore not independent from each other (Dommain
et al., 2020; Mitchell et al., 2005; Wakeham and Canuel, 2006; Zonneveld et al., 2010).
Evolutionary models require knowledge of how rapidly and how far the aforementioned
traits can change from generation to generation, as well as of the trade-offs between traits
(Levins, 1962; Litchman et al., 2007) and ultimate constraints on adaptation (Klausmeier et al.,
2020a). Such information is available from evolution experiments (Hinners et al., 2024), but it
is still unclear how applicable such information will be when moving from a highly simplified
evolution experiment to a more complex community context. A major challenge is to link trait
changes to changes in fitness. While the relationship between a fitness proxy and actual fitness
may change over time (Collins et al., 2020), fitness is largely determined by species interactions
(Schabhüttl et al., 2013). Based on the assumption that the fittest phytoplankton taxa are also
the most abundant in the sediment, sediment archives make it possible to estimate the relative
fitness of different taxa. However, the validity of this assumption might be challenged by

ecological and evolutionary complexities, and possible preservation differences between taxa. Preservation differences might be evaluated by synthesizing information from different proxies of abundance (e.g., biomarkers and sedimentary ancient DNA). Knowledge on species' biology and ecology should be informative about the confounding effects of ecological and evolutionary complexities, such as demographic trade-offs, differential responses to environmental fluctuations (Melbinger and Vergassola, 2015), temporal trade-offs (Betini et al., 2017), etc. Consequently, the validity of the fitness = abundance assumption should be tested in each individual case.

Despite limitations and knowledge gaps, sediment archives represent a valuable source of information that has the potential to advance ecosystem model development and hence model projections of marine ecosystem change. As pointed out above, a crucial step in ecosystem model development is to make sure that models are equally applicable to past, present, and future scenarios before using them predictively. This requires validation data that are independent of the data used for calibration. Moreover, validation data need to cover the complexity of marine ecosystems and long-term environmental changes over hundreds to thousands of years. While data from laboratory, mesocosm, or marine monitoring studies only partly fulfill these criteria, sediment archives fulfill all of them. Furthermore, the approach presented here is not limited to phytoplankton, but can be applied to other organisms that are well-represented in the sediment record, such as marine microbes (Wörmer et al., 2019), zooplankton (Isanta-Navarro et al., 2021; Wersebe and Weider, 2023), viruses (Coolen, 2011), and terrestrial plants (Alsos et al., 2024). Depending on the chosen group of organisms, its applicability for evolution experiments, and its potential to survive in the sediment or to be identified by microfossils or DNA, our described approach may require adjustment.

# 5 Conclusions

Marine communities perform functions that are essential for the environment and for humanity. However, it is largely unknown how these functions will change under global warming, and the possibility of positive feedbacks, irreversible tipping points, and ecosystem collapse must be considered. It is therefore crucial to develop tools that provide reliable estimates of future changes in marine ecosystems.

Ecosystem models represent a promising tool for predicting marine ecosystem change,
but their current projections are largely inconsistent. Here, we present a promising approach
that can increase the informative value of ecosystem model projections. We argue that a major
uncertainty in current ecosystem models is that they largely ignore evolutionary processes,
which can be highly relevant on perennial time scales. In addition, current ecosystem models
are typically calibrated to contemporary data and then used for projections without validating
that they are equally applicable to past (and by implication, future) scenarios. We suggest not
only to calibrate evolutionary ecosystem models against contemporary observations, but also
to validate the calibrated models against major evolutionary ecosystem changes that are present
in sediment records. Compared to data from conventional experiments and marine monitoring,
sediment records make it possible to map the complexity of real ecosystems and long-term
environmental changes. Only if a contemporarily calibrated evolutionary ecosystem model can
reproduce observations from sediment records, can we have some confidence in its projections
of future ecosystem change.
Some challenges remain, especially regarding the low temporal resolution of sediment
archives and potential biases due to preservation differences. Nevertheless, data from sediment
archives provide a unique opportunity to learn from the past and hence have the potential to
take ecosystem models and their projections of future ecosystem change a crucial step forward.
The approach presented here is not limited to phytoplankton, but can be applied to other
organisms and ecosystems.

# Author contribution

**Isabell Hochfeld:** Conceptualization; project administration; visualization; writing – original
draft; writing – review and editing. **Ben A. Ward:** Conceptualization; writing - original draft;
writing – review and editing. **Anke Kremp:** Conceptualization; writing – original draft; writing
– review and editing. **Juliane Romahn:** Conceptualization; visualization; writing – original
draft; writing – review and editing. **Alexandra Schmidt:** Conceptualization; visualization;
writing – original draft; writing – review and editing. **Miklós Bálint:** Conceptualization;
writing – original draft; writing – review and editing. **Lutz Becks:** Conceptualization; writing
– original draft; writing – review and editing. **Jérôme Kaiser:** Conceptualization; writing –
original draft; writing – review and editing. **Helge W. Arz:** Conceptualization; writing – review

and editing. **Sarah Bolius:** Conceptualization; writing – review and editing. **Laura S. Epp:** Conceptualization; writing – review and editing. **Markus Pfenninger:** Conceptualization; writing – review and editing. **Christopher A. Klausmeier:** Conceptualization; writing – review and editing. **Elena Litchman:** Conceptualization; writing – review and editing. **Jana Hinners:** Conceptualization; project administration; writing - original draft; writing – review and editing.

# Competing interests

The authors declare that they have no conflict of interest.

# Acknowledgements

This work was made possible by the funding of the project PhytoArk (K314/2020) by the Leibniz Association. BAW was funded by a Royal Society University Research Fellowship. EL and CAK acknowledge US NSF grants 17-54250 and 21-24800. JR, MB, and MP acknowledge support by the LOEWE-Centre TBG funded by the Hessen State Ministry of Higher Education, Research, and the Arts (HMWK, grant number LOEWE/1/10/519/03/03.001(0014)/52).

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

H. C., Lappalainen, T., and Posthuma, D.: Genome-wide association studies, Nature Reviews
Methods Primers, 1, 59, 2021.
Umlauf, L., Burchard, H., and Bolding, K.: GOTM-scientific documentation: version 3.2,
Institut für Ostseeforschung, 2005.
Van Nieuwenhove, N., Head, M. J., Limoges, A., Pospelova, V., Mertens, K. N.,
Matthiessen, J., De Schepper, S., de Vernal, A., Eynaud, F., Londeix, L., and others: An
overview and brief description of common marine organic-walled dinoflagellate cyst taxa
occurring in surface sediments of the Northern Hemisphere, Marine Micropaleontology, 159,
1068 101814, 2020.

Vasselon, V., Bouchez, A., Rimet, F., Jacquet, S., Trobajo, R., Corniquel, M., Tapolczai, K.,
and Domaizon, I.: Avoiding quantification bias in metabarcoding: Application of a cell
biovolume correction factor in diatom molecular biomonitoring, Methods in Ecology and
Evolution, 9, 1060–1069, 2018.
Vincent, W. and Silvester, W.: Growth of blue-green algae in the Manukau (New Zealand)
oxidation ponds—I. Growth potential of oxidation pond water and comparative optima for
blue-green and green algal growth, Water Research, 13, 711–716, 1979.
Visscher, P. M., Wray, N. R., Zhang, Q., Sklar, P., McCarthy, M. I., Brown, M. A., and
Yang, J.: 10 years of GWAS discovery: biology, function, and translation, The American
Journal of Human Genetics, 101, 5–22, 2017.
Wakeham, S. G. and Canuel, E. A.: Degradation and preservation of organic matter in marine
sediments, Marine Organic Matter: Biomarkers, Isotopes and DNA, 295–321, 2006.
Ward, B. A., Dutkiewicz, S., Jahn, O., and Follows, M. J.: A size-structured food-web model
for the global ocean, Limnol. Oceanogr., 57, 1877–1891, 2012.

Ward, B. A., Collins, S., Dutkiewicz, S., Gibbs, S., Bown, P., Ridgwell, A., Sauterey, B., Wilson, J., and Oschlies, A.: Considering the role of adaptive evolution in models of the ocean and climate system, J. Adv. Model. Earth Syst., 11, 3343–3361, 2019.

Wasmund, N., Nausch, G., Gerth, M., Busch, S., Burmeister, C., Hansen, R., and Sadkowiak, B.: Extension of the growing season of phytoplankton in the western Baltic Sea in response to climate change, Mar. Ecol. Prog. Ser., 622, 1–16, 2019.

Weckström, K.: Assessing recent eutrophication in coastal waters of the Gulf of Finland (Baltic Sea) using subfossil diatoms, Journal of Paleolimnology, 35, 571–592, 2006.

Weitz, J. S., Stock, C. A., Wilhelm, S. W., Bourouiba, L., Coleman, M. L., Buchan, A., Follows, M. J., Fuhrman, J. A., Jover, L. F., Lennon, J. T., Middelboe, M., Sonderegger, D. L., Suttle, C. A., Taylor, B. P., Frede Thingstad, T., Wilson, W. H., and Eric Wommack, K.: A multitrophic model to quantify the effects of marine viruses on microbial food webs and ecosystem processes, ISME J, 9, 1352–1364, https://doi.org/10.1038/ismej.2014.220, 2015.

Wersebe, M. J. and Weider, L. J.: Resurrection genomics provides molecular and phenotypic evidence of rapid adaptation to salinization in a keystone aquatic species, Proceedings of the National Academy of Sciences, 120, e2217276120, 2023.

Willerslev, E., Hansen, A. J., Binladen, J., Brand, T. B., Gilbert, M. T. P., Shapiro, B., Bunce, M., Wiuf, C., Gilichinsky, D. A., and Cooper, A.: Diverse plant and animal genetic records from Holocene and Pleistocene sediments, Science, 300, 791–795, 2003.

Wittenborn, A. K., Radtke, H., Dutheil, C., Arz, H. W., and Kaiser, J.: A downcore calibration of the TEX86L temperature proxy for the Baltic Sea, Continental Shelf Research, 251, 104875, 2022.

Wood, S. M., Kremp, A., Savela, H., Akter, S., Vartti, V.-P., Saarni, S., and Suikkanen, S.: Cyanobacterial akinete distribution, viability, and cyanotoxin records in sediment archives from the Northern Baltic Sea, Frontiers in Microbiology, 12, 681881, 2021.

Wörmer, L., Hoshino, T., Bowles, M. W., Viehweger, B., Adhikari, R. R., Xiao, N., Uramoto, G., Könneke, M., Lazar, C. S., Morono, Y., and others: Microbial dormancy in the marine subsurface: global endospore abundance and response to burial, Science Advances, 5, eaav1024, 2019.

Zimmermann, H. H., Stoof-Leichsenring, K. R., Dinkel, V., Harms, L., Schulte, L., Hütt, M.-
T., Nürnberg, D., Tiedemann, R., and Herzschuh, U.: Marine ecosystem shifts with deglacial
sea-ice loss inferred from ancient DNA shotgun sequencing, Nature Communications, 14,

1115     1650, 2023.

Zonneveld, K. A., Versteegh, G. J., Kasten, S., Eglinton, T. I., Emeis, K.-C., Huguet, C.,
Koch, B. P., de Lange, G. J., de Leeuw, J. W., Middelburg, J. J., and others: Selective
preservation of organic matter in marine environments; processes and impact on the
sedimentary record, Biogeosciences, 7, 483–511, 2010.