# Peer review of "Ideas and perspectives: How sediment archives can"

_EGUsphere, 2024_

## Referee Comment (RC1)

Review of Hochfeld et al, "How sediment archives can improve model projections of marine ecosystem change"

**Summary**

This manuscript is a perspective piece outlining an approach to validate evolutionary ecosystem models using sediment archives. The authors discuss the limitations of current models and the advantages of using long-term integrated records such as sediments to constrain evolutionary changes. They introduce a framework in which this can be applied and discuss remaining challenges.

**General comments**

The main concept of the paper is well articulated and convincing. However, the actual application of the framework to real models and archives is not as clear. I suggest adding a section that contains a more quantitative, worked example with a specific model and sedimentary record, including details such as what model structural aspects and/or parameters were considered for calibration, what proxies were used to constrain the model, what adjustments were made upon inclusion of the proxy data, and how much it reduced uncertainty.

I would also like to see a more robust discussion of uncertainties in both models and proxies. For example, how does the age model uncertainty and temporal resolution of sediment archives affect derived rates of evolutionary change? Do different rates of preservation under different environmental conditions limit the conclusions you can make about relative abundance and evolutionary change?

A definition of precisely what you mean by "evolutionary ecosystem model" would also be helpful (and contrast with ecosystems that are not evolutionary). I suggest including a conceptual diagram of how these models generally work.

**Specific comments**

L26: Include more recent references for projected changes in marine ecosystems e.g. CMIP6 Fish-MIP results: Tittensor, D.P., Novaglio, C., Harrison, C.S. *et al. Nat. Clim. Chang.* **11**, 973–981 (2021). https://doi.org/10.1038/s41558-021-01173-9

L41: be more specific about "positive feedbacks" and "tipping points"

L46-48: "Since models hardly agree ..." This statement is overly simplistic; the validity of model projections depends on much more than inter-model agreement, for example model complexity, ability to match present-day observations, etc. It is the ability to

evaluate *which* model projections are valid that is the challenge. Give specific numbers of the range of predicted change.

L49-71: This paragraph would be a good place to discuss different types and complexities of ecosystem models and what is meant by evolutionary adaptation in this context

L49-50: " ... to verify that all relevant processes are considered ..." This is extremely unlikely with current models and computing capacity; a better statement would be "*the most* relevant processes"

L60: "... allow  the reconstruction of long-term ..."

L68-69: what time resolution is needed to constrain evolutionary changes?

Section 2.1: add discussion of age model uncertainties (particularly marine reservoir ages for Radiocarbon) and how this could impact the ability to validate models with sediment archives

L117: "... allow  the reconstruction of ..."

L151: Don't begin a sentence with "also"

L161: "resting stage" should be described and/or defined when this term is introduced (also in Figure 1 caption)

L186 "This approach potentially allows one to obtain ..."

L203: " ... and the mechanisms behind the changes"

L209: How would you derive grazing rates from these experiments?

L245: define "perennial time scales"

Section 3.1: This section could be expanded to talk more about different kinds of evolutionary models and how they integrate adaptation

L261: What are "evolution experiments"?

L330-331: be more specific about the location of the 5 deg C temperature anomaly during the Holocene thermal maximum, e.g. high latitudes rather than global. Why not other past warm intervals, such as the Last Interglacial or the Pliocene, which are potentially more extreme?

L345: How do you know that the problem is structural and not just due to a few parameters that are incorrect? This example needs more detail

L389-390: Assuming that the fittest individuals are the most abundant is questionable; "fitness" is an abstract concept and rates of preservation in the sediment archive can heavily bias derived abundance rates. How might this affect your conclusions?

---

## Author Comment (AC1)

**Reply to Review 1**

We would like to thank the anonymous reviewer very much for their helpful and constructive feedback, and we would be happy to revise our manuscript by accounting for their valuable points of criticism.

1) The reviewer would like to see a more quantitative, worked example with a specific model and sedimentary record. We understand that such an example would make it clearer how we envisage the actual application of our approach. However, we designed this manuscript as an "Ideas and perspectives" article with the purpose to propose a new concept for ecosystem model development that can be tested, refined, and applied by future research. We believe that a complete working example is beyond the scope of this article. However, we suggest refining the explanation of the actual application of our approach, e.g., by providing concrete examples of the different aspects of our proposed concept.

2) We suggest revising and extending our discussion of uncertainties in both models and proxies as recommended by the reviewer. In the revised version of our discussion, we will focus more on the effects of age model uncertainty, temporal resolution, and preservation biases of sedimentary archives on estimated rates of evolutionary change and relative abundance.

3) We agree that a clearer definition of what evolutionary ecosystem models are, as opposed to non-evolutionary ones, would greatly help our manuscript. Since many different approaches exist to include evolutionary processes into ecosystem models, it would be difficult to visualize them in a single diagram. Therefore, we suggest adding a text box to our manuscript, in which we explain the general idea of simulating evolution in ecosystem models, along with different state-of-the-art approaches.

4) Finally, we will address all the specific comments raised by the reviewer.

---

## Author Comment (AC2)

**Reply to Review 2**

We would like to thank Damien Eveillard very much for his constructive comments and helpful advice on our manuscript. We would be happy to account for his points of criticism in a revised manuscript version.

1) As suggested, we will point out more clearly the constraints arising when applying our approach to other biological entities, such as heterotrophic bacterial populations.

2) Unfortunately, we didn't quite understand what the reviewer means by discussing model plasticity. Based on our understanding, the term plasticity refers to the physiological plasticity of organisms as a response to altered environmental conditions. This definition does not align with the reviewer's comment, though. However, we hope that our revised version will account for this criticism.

3) We will add a more extensive discussion on the potential risk of model overfitting.

4) We agree that our manuscript would benefit from a thorough presentation of contemporary evolutionary modeling approaches. As also suggested by Reviewer 1, we think that a text box would be appropriate to explain different state-of-the-art modeling approaches. In that box, we can further clarify the rationale behind these approaches and evolutionary ecosystem models in general (i.e., the underlying mechanisms in the form of mathematical equations and parameters). This would also allow us to explain how both new and historical data can be incorporated into these models and how changes in ecosystem structure can be simulated.

5) As suggested, we will make a stronger link between section 2.3.3 and the rest of the manuscript and expand lines 206-210 regarding phytoplankton traits that are of particular relevance for ecosystem modeling. We are not completely sure what the reviewer means by asking for approaches to connect GWAS analysis and identified characteristics. However, we suggest to explain that it is necessary to validate the candidate loci identified in GWAS by experiments that target the phenotypic functionality of these loci.

6) While we like the idea of connecting figures 1 & 2 by the same color-coding, we see major practical complications, since both figures have a different level of detail. In Fig. 1, we distinguish between different sources of sedimentary data (sedimentary ancient DNA, biomarkers, microfossils…), whereas in Fig. 2, we focus on how information from different sources can be integrated into ecosystem models. As an example, information on biodiversity can be obtained from different data sources, such as sedimentary DNA, microfossils, and biomarkers, making a simple color-coding impossible. Therefore, it will unfortunately not be possible to establish a color code for these two figures. However, we will still try to establish a better connection between

both figures through adjusting the wording (e.g., by replacing "resurrection experiments" in Fig. 2 with resurrectable resting stages).

7) We will further expand on how DNA sediment data may limit quantitative parameter assessments.

8) Finally, we agree that we should further clarify the difference between model structure and parameters. We think that the text box we proposed above would be a good opportunity for that.

---

## Author Response (AR1)

**Author's response:**

Dear Sebastian Naeher,

Thank you very much for the clear suggestions to revise our manuscript. We have now revised our manuscript to address all of the reviewers' comments. Below, we explain the changes we have made to our manuscript with respect to each reviewer's comment. Besides many minor changes, the two major changes are a text box explaining evolutionary ecosystem models and an exemplary implementation of our approach. We look forward to hearing from you and thank you on behalf of all co-authors.

Kind regards,

Jana Hinners and Isabell Hochfeld

**Review 1:**

Summary
This manuscript is a perspective piece outlining an approach to validate evolutionary ecosystem models using sediment archives. The authors discuss the limitations of current models and the advantages of using long-term integrated records such as sediments to constrain evolutionary changes. They introduce a framework in which this can be applied and discuss remaining challenges.

General comments
The main concept of the paper is well articulated and convincing. However, the actual application of the framework to real models and archives is not as clear. I suggest adding a section that contains a more quantitative, worked example with a specific model and sedimentary record, including details such as what model structural aspects and/or parameters were considered for calibration, what proxies were used to constrain the model, what adjustments were made upon inclusion of the proxy data, and how much it reduced uncertainty.
We have incorporated an exemplary implementation into the revised version (L501-563).

I would also like to see a more robust discussion of uncertainties in both models and proxies. For example, how does the age model uncertainty and temporal resolution of sediment archives affect derived rates of evolutionary change? Do different rates of preservation under different environmental conditions limit the conclusions you can make about relative abundance and evolutionary change?
We have added a detailed discussion of age model uncertainty, temporal resolution of sediment archives, and preservation, including their potential impact on estimated evolutionary rates and relative abundances (L581-600).

A definition of precisely what you mean by "evolutionary ecosystem model" would also be helpful (and contrast with ecosystems that are not evolutionary). I suggest including a conceptual diagram of how these models generally work.
We have included a text box (Box 1) and a table (Table 1) to explain what evolutionary ecosystem models are and what different types of models exist (L340-391).

Specific comments
L26: Include more recent references for projected changes in marine ecosystems e.g.

CMIP6 Fish-MIP results: Tittensor, D.P., Novaglio, C., Harrison, C.S. et al. Nat. Clim. Chang. 11, 973–981 (2021). https://doi.org/10.1038/s41558-021-01173-9
Changed accordingly (L67, 92, 94, 309-310)

L41: be more specific about "positive feedbacks" and "tipping points"
We have extended the explanation (L82-86).

L46-48: "Since models hardly agree …" This statement is overly simplistic; the validity of model projections depends on much more than inter-model agreement, for example model complexity, ability to match present-day observations, etc. It is the ability to evaluate which model projections are valid that is the challenge. Give specific numbers of the range of predicted change.
Changed accordingly (L91-95)

L49-71: This paragraph would be a good place to discuss different types and complexities of ecosystem models and what is meant by evolutionary adaptation in this context
We have decided to include a text box further down in the manuscript to give this topic more space than would be possible in the introduction (see L340-391).

L49-50: " … to verify that all relevant processes are considered …" This is extremely unlikely with current models and computing capacity; a better statement would be "the most relevant processes"
Changed accordingly (L98)

L60: "…  the reconstruction of long-term …"
Changed accordingly (L109)

L68-69: what time resolution is needed to constrain evolutionary changes?
This is discussed in more detail now in the discussion section (L581-600).

Section 2.1: add discussion of age model uncertainties (particularly marine reservoir ages for Radiocarbon) and how this could impact the ability to validate models with sediment archives
See extended discussion in (L581-600).

L117: "… allow the reconstruction of …"
Changed accordingly (L171-172)

L151: Don't begin a sentence with "also"
Changed accordingly (L205-206)

L161: "resting stage" should be described and/or defined when this term is introduced (also in Figure 1 caption)
Changed accordingly (L113-116 and L145-146)

L186 "This approach potentially allows one to obtain …"
Changed accordingly (L240)

L203: " … and the mechanisms behind the changes"
Changed accordingly (L260)

L209: How would you derive grazing rates from these experiments?
Based on a comment from the other reviewer, we have changed this section and removed grazing rates (L260-274).

L245: define "perennial time scales"
Changed accordingly (L313)

Section 3.1: This section could be expanded to talk more about different kinds of evolutionary models and how they integrate adaptation
We have included a text box and a table (Box 1 and Table 1) into this section explaining evolutionary ecosystem models (L340-391).

L261: What are "evolution experiments"?
We have added an explanation (L332-335).

L330-331: be more specific about the location of the 5 deg C temperature anomaly during the Holocene thermal maximum, e.g. high latitudes rather than global. Why not other past warm intervals, such as the Last Interglacial or the Pliocene, which are potentially more extreme?
We have modified this part accordingly (L462-469).

L345: How do you know that the problem is structural and not just due to a few parameters that are incorrect? This example needs more detail
The reviewer is correct, we do not know that the problem is structural. We have changed the text to reflect this: "This indicates that the model would require further adjustments to allow reproduction of both contemporary and sedimentary data before the model could be used predictively" (L484-486).

L389-390: Assuming that the fittest individuals are the most abundant is questionable; "fitness" is an abstract concept and rates of preservation in the sediment archive can heavily bias derived abundance rates. How might this affect your conclusions?
This is indeed an important consideration that we have now included and discussed (L606-607, L623-631).

**Review 2:**

Hochfeld and colleagues promote the utilization of sediment archives to enhance modeling efforts. Specifically, this thesis supports the integration of evolutionary processes, which require consideration of extended temporal scales that are typically unattainable through conventional longitudinal datasets, such as mesocosms. By incorporating sediment archives, researchers can gain valuable insights into historical ecological dynamics and better understand the long-term impacts of environmental changes on species evolution and community structure.
The manuscript is informative and emphasizes their hypothesis in the context of phytoplankton, evaluates the prevailing literature, and elucidates the potential insights that may be derived from sediment archives. It provides an exposition of various datasets and delineates the methodologies through which diverse knowledge can be obtained. Furthermore, it articulates a modeling pipeline designed to enhance the synthesis of sediment archive information, thereby facilitating improved predictive outcomes. This comprehensive approach highlights the significance of sediment archives in ecological research and paves the way for future studies to integrate these valuable datasets into broader environmental assessments.

Specific comments:

It is frequently noted that although the primary emphasis is on phytoplankton, the methodological framework could be applied to other biological entities. Incorporating the constraints associated with each dataset concerning alternative organisms may be beneficial. For example, the implications of microfossils may not be extendable to heterotrophic bacterial populations. Incorporating these constraints will provide a more nuanced understanding of the ecological dynamics at play, ensuring that researchers can accurately interpret the data and its relevance to different biological communities. Similarly, I recommend enhancing the limitations for (heterotrophic) prokaryotic systems, as following the Falkowski paradigm (i.e., microbial engines), these organisms are central for climate mitigations. Expanding the scope to include limitations will enhance the comprehensiveness of the assessments, allowing for a more holistic view of microbial interactions and their implications in various ecosystems.

We have added marine microbes to the list of organism groups to which our approach could be applied (L246, 642). Although important, an extensive discussion of the applicability and limitations for these other organism groups would go beyond the scope of this paper. Instead, we have mentioned that our approach would need adjustment depending on the chosen organism group (L644-646).

On another point, I would recommend adding a more extensive discussion about model plasticity. Plasticity could take the form of community structure modification, better data fitting, and new modeling techniques that could better embed more diverse data (?) with the potential risk of overfitting.

We have added more information on how model parameters and structure can be adjusted (L447-452), and describe how model uncertainties can be estimated (L496-499).

The manuscript would gain considerable value from including a more formalized definition of adaptation, particularly concerning contemporary modeling techniques. Specifically, it would be beneficial to incorporate an additional figure elucidating the rationale underlying this concept, which is frequently overlooked. Furthermore, section 3 would significantly enhance its clarity by providing a more adequate representation of the rationale behind models of evolutionary ecosystems. This could be achieved by including a box or a figure panel illustrating where (e.g., equations or parameter values) the model stands to gain from integrating both new and historical data. Moreover, it is essential to articulate how changes in ecosystem structure can be formalized. This would facilitate a deeper understanding of the model's dynamics and highlight the importance of adaptive management strategies in response to environmental shifts.

We have added a text box and a table (Box 1 and Table 1) to explain different concepts of evolutionary ecosystem models (L340-391). However, we have not extended our discussion towards management strategies since that would reach far beyond the scope of this paper.

Section 2.3.3, entitled "Resurrection Experiments," presents an intriguing perspective; however, it does not seamlessly integrate with the overall narrative of the manuscript. To mitigate this disruption in the continuity of the storyline, would it be possible to elaborate on the methodology for integrating this supplementary knowledge into the modeling discussions? In particular, could lines 206-210 further expand the content? Additionally, what approaches could be employed to establish a connection between GWAS analysis and the identified characteristics (i.e., growth traits)?

We have changed the title of this section to "Resurrectable dormant resting stages" and extended the information in the respective lines (now L260-274). We have also added more context to the GWAS analyses (L279-280).

In Figure 2, I would add a color code to the data that matches the one in Figure 1. This is a cosmetic proposition, but it would make the figures more connected, cohesive, and visually appealing.

Unfortunately, we have not been able to match the color coding because the level of detail is not the same, but we have aligned the terms as much as possible, changing "resurrection experiments" to "resurrectable resting stages".

DNA sediment data, like DNA itself, is compositional in nature. Mixing such data with quantitative measurements for parameter assessments introduces limitations, which the authors should explicitly acknowledge.

We have added information on this (L286-288).

The distinction between model structure and parameter values could be elaborated further for clarity. A more detailed explanation would help readers understand the conceptual differences, as the proposed modeling pipeline figure covers mainly parameterization issues.

We have added a differentiated explanation of how model parameters and structure can be adjusted (L447-452), and how their uncertainties can be estimated (L496-499).